# ObjBlur: A Curriculum Learning Approach With Progressive Object-Level Blurring for Improved Layout-to-Image Generation

## ABSTRACT

We present ObjBlur, a novel curriculum learning approach to improve layout-to-image generation models, where the task is to produce realistic images from layouts composed of boxes and labels. Our method is based on progressive object-level blurring, which effectively stabilizes training and enhances the quality of generated images. This curriculum learning strategy systematically applies varying degrees of blurring to individual objects or the background during training, starting from strong blurring to progressively cleaner images. Our findings reveal that this approach yields significant performance improvements, stabilized training, smoother convergence, and reduced variance between multiple runs. Moreover, our technique demonstrates its versatility by being compatible with generative adversarial networks and diffusion models, underlining its applicability across various generative modeling paradigms. With ObjBlur, we reach new state-of-the-art results on the complex COCO and Visual Genome datasets.

## CCS CONCEPTS

• **Computing methodologies → Neural networks**; **Computational photography**; **Image processing**.

## KEYWORDS

Image Generation, Curriculum Learning, Layout-to-Image

## 1 INTRODUCTION

Layout-to-image generation is a fundamental task in computer vision and graphics, bridging the gap between structured scene descriptions, such as layouts composed of bounding boxes and labels, and the generation of realistic images [10, 29, 39, 43]. It is a complex task, further compounded by intrinsic variations in the difficulty of learning to generate different object classes and their inherent diversity in shapes, sizes, and context [8].

Layout-to-image models are mainly based on GANs [9] and thus inherit their training stability issues, such as mode collapse and overfitting [22]. While data augmentation (DA) techniques have been proven to be effective in visual recognition models [25, 35], training a GAN under similar augmentations leads to a leaking effect in which the generator learns to produce augmented (instead of clean) images. For example, if rotation is used as a DA, the generator will produce rotated images after training, an undesirable

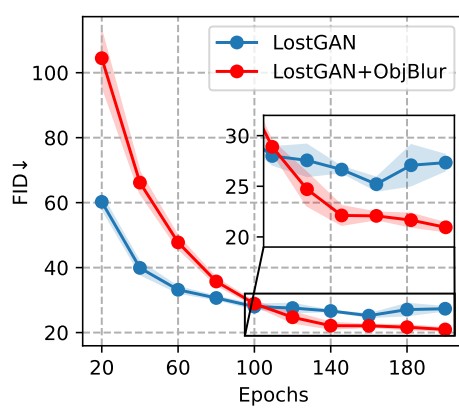

**Figure 1: Comparison of FID during training. ObjBlur stabilizes training, leading to smoother convergence with better final performance and lower standard deviation across three runs, especially at the end of training.**

outcome. To mitigate this problem, consistency regularization [37, 41, 42], invertibility [31], and differential augmentation techniques [15, 40] have been proposed.

Meanwhile, the machine learning community has been interested in curriculum learning (CL) strategies [1, 28, 32] to structure training examples in a meaningful order that gradually exposes the model to more complex concepts. It provides an intuitive approach to guiding models through progressively challenging training scenarios. Interestingly, their exploration remains relatively limited in the context of generative models [28, 32] and nonexistent in the domain of layout-to-image generation. For single-object generative image models, previous work proposed the use of multiple discriminators [7, 13, 24], progressively growing the model [14] or ranking images by difficulty [27]. However, all previous work requires either changing the model, loss function, using a difficulty estimator, or a combination of them. To our knowledge, there is no previous work on using curriculum learning for layout-to-image generation.

This paper introduces ObjBlur, a new approach to layout-to-image generation that utilizes curriculum learning by applying progressive object-level blurring to improve the image quality of layout-to-image models. Blurring is a natural image degradation operation because low frequencies are retained over higher frequencies. In fact, even human perception is more sensitive to low frequencies of an image [2, 23]. Strong blurring removes high-frequency details, resulting in a simpler signal without affecting the structural content of the image (as opposed to degradation alternatives such as additive noise). Decreasing the blur strength produces a more complex signal with high-frequency details, thus exposing the model to a more difficult task. Therefore, blurring

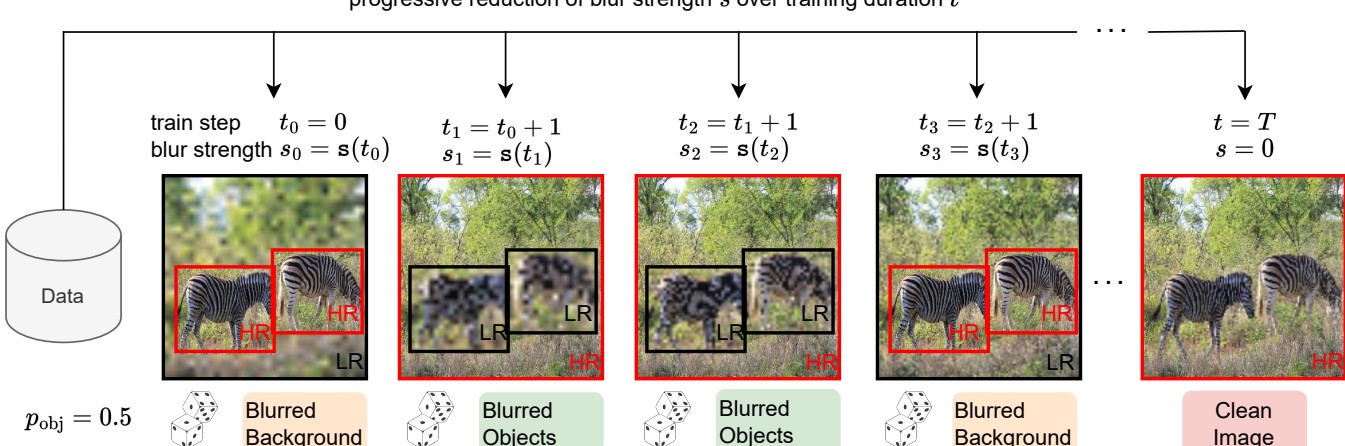

**Figure 2: Our ObjBlur method incorporates a novel curriculum learning approach based on progressive object-level blurring to individual objects or the background throughout the training procedure on a per-sample basis. At each training step $t_i$, we use the blurring schedule function $\mathsf{s}(t_i)$ to compute the current blurring strength $s_i$, starting from strong blurring to progressively cleaner images. Finally, the probability $p_{\mathbf{obj}}$ defines whether blurring should be applied to objects or the background for the current image. More details in section 3.**

offers an intuitive and powerful approach to incrementally adjust task difficulty, ensuring a smooth training progression.

Our method can be realized by only modifying the data loader to apply a progressive blur to the images. As a result, it can be easily integrated into existing layout-to-image approaches and does not depend on difficulty estimators or changes in the model architecture and optimization protocol. By systematically applying varying degrees of blurring during training, starting with strong blurring and progressing to cleaner images, we stabilize training and ensure that the model learns to generate high-quality images.

A crucial aspect of image quality is the appearance of foreground objects in relation to the background. Thus, we propose an object-level approach that randomly applies the blur to either the objects or the background. To demonstrate the benefits of ObjBlur, we perform extensive analysis on several layout-to-image generation models, including adversarial- [10, 29], and diffusion-based [43] approaches. We also comprehensively analyze several design choices and their impact on performance and stability. Using LayoutDiffusion [43] as a backbone, our proposed ObjBlur schedule significantly improves the quality of generated images, offering a robust and versatile approach that leads to new state-of-the-art results. In terms of FID [11], SceneFID [30] and CAS [20], we reach relative improvements of 2.38%, 34.43%, 4.70% on COCO [3], and 6.13%, 18.45%, 10.15% on Visual Genome [17] while only requiring changes to the dataloader.

## 2 RELATED WORK

### 2.1 Layout-to-Image

The layout-to-image (L2I) task was first studied in [39] using a VAE [16] by composing object representations into a scene before producing an image. Adversarial approaches [29] produced higher-resolution images and provided better control of individual objects

by using a reconfigurable layout with separate latent style codes. Further developments studied better instance representations [30] and context awareness [10]. Recent developments have focused on using diffusion models by adjusting the self-attention mask to focus on the instances and adding prompt tokens [5] or by constructing a structural image patch with region information to facilitate multi-modal fusion of image and layout [43].

### 2.2 Curriculum Learning

The idea of monotonically increasing the difficulty of tasks is related to curriculum learning (CL) [1] which introduces more complex concepts gradually, instead of randomly presenting training data. CL is inspired by the teaching paradigm of organizing learning material in an orderly fashion. Although it has been successfully applied in a wide range of tasks [28, 32], its application to generative models is minimal. In text generation, the length of character sequences can gradually be increased as training progresses [19]. To improve image generation, multiple discriminators are used in [7, 13, 24], while the image resolution increases in [14]. While [14] is similar in spirit, it requires an entirely different implementation to grow both the generator and discriminator layers during training, a challenging and not generalizable procedure. In [4], a CL strategy based on semantic difficulty determined by embedding distance is used for text-to-image synthesis. Several CL strategies are proposed in [27] based on ranking the training images according to their difficulty scores, and a smoothing schedule to intermediate CNN features is presented in [26]. To the best of our knowledge, we propose the first CL strategy for L2I models using a progressive object-level blurring schedule without requiring architectural changes to the model. Instead, our method only requires dataloader changes and can thus be seamlessly integrated into any method.

## 2.3 Blurring & other DA Techniques

Data augmentation (DA) techniques have played an essential role in the success of deep learning over the last decade by artificially expanding the data set and enabling the training of large models [25, 35]. Examples of model-free image augmentation can be categorized into geometrical transformations, color space augmentations, kernel filters, image and feature mixing, and random erasing. Unfortunately, training generative models under similar augmentations typically leads to a leaking effect where the generator learns to produce the augmented data distribution. Our method is inspired by CutBlur [36], a data augmentation technique developed to improve the performance of super-resolution models. CutBlur pastes a low-resolution patch into the corresponding high-resolution image region and vice versa. On the contrary, our method is tailored to generate images from layouts and is a creative application of CL. It uses progressive object-level blurring as a CL strategy and solves the leaking issue by progressing from strong blur to clean images during training, thus guiding our model to produce clean images at inference time.

## 2.4 Diffusion GANs

Diffusion models achieve excellent sample quality but traditionally suffer from expensive sampling. GANs [9] can generate high-quality images in one step but are typically prone to training instability, such as overfitting and mode collapse [22]. Recently, approaches combining GANs with diffusion models have become popular to address this issue. In [34], the diffusion model is parameterized by a multimodal conditional GAN to smoothen the data distribution and increase the sampling step size. In [33], instance noise is injected using a diffusion timestep-dependent discriminator to stabilize training, augment the dataset, and ease the vanishing gradient problem. In contrast, we propose a CL strategy using an object-level blurring schedule during training to improve layout-to-image models.

## 2.5 Blurring Diffusion Models

An interesting connection, especially when combining our idea with diffusion models, can be made with recent heat dissipation [21] and blurring diffusion [6, 12, 18] models. In [21], images are generated by stochastically reversing the heat equation, corresponding to a blur operator. Meanwhile, [12] defines blurring through a diffusion process with non-isotropic noise, combining heat dissipation and additive noise. In [18], each frequency component of an image is diffused at different speeds, resulting in a reverse process that gradually deblurs and removes noise. The pairing of blur with noise as the diffusion mechanism is also proposed in [6]. In contrast to previous research, we maintain the diffusion and sampling processes unchanged. Instead, we adapt the data loader to implement a curriculum learning strategy, facilitating a seamless transition from an easier to a more challenging task.

## 3 METHOD

This section describes our curriculum learning strategy based on progressive object-level blurring using notation similar to that in CutBlur [36]. Our ObjBlur method is straightforward: Given a clean, high-resolution image $\mathbf{x}_{HR} \in \mathbb{R}^{W \times H \times C}$ and layout $\ell = \{(b_i, c_i)_{i=1}^m\}$

---

**Algorithm 1** ObjBlur

**Input:** $\mathcal{D}$: training dataset
**Input:** $M_\theta$: initialized model
**Input:** $p_{obj}$: object blur probability
**Input:** $s(t)$: blur schedule function
1: **for all** training steps $t = 0$ to $T$ **do**
2:     $\mathbf{x}_{HR}, \ell \sim \mathcal{D}$      ▷ Sample image and layout
3:     $s_t = s(t)$      ▷ Get blur strength
4:     $\mathbf{x}_{LR} = \text{blur}(\mathbf{x}_{HR}, s_t)$      ▷ Get LR image
5:     $\mathbf{m} = \text{binarize}(\ell)$      ▷ Get binary mask
6:     **if** $p_{obj} \leq \mathcal{U}(0, 1)$ **then**
7:         $\hat{\mathbf{x}} = \mathbf{m} \odot \mathbf{x}_{LR} + \overline{\mathbf{m}} \odot \mathbf{x}_{HR}$      ▷ Blur objects
8:     **else**
9:         $\hat{\mathbf{x}} = \mathbf{m} \odot \mathbf{x}_{HR} + \overline{\mathbf{m}} \odot \mathbf{x}_{LR}$      ▷ Blur background
10:     **end if**
11:     $M_\theta \leftarrow \text{step}(M_\theta; \hat{\mathbf{x}}, \ell)$      ▷ Train step using $\hat{\mathbf{x}}$ and $\ell$
12: **end for**
**Output:** Trained model $M_\theta$

---

of $m$ objects with corresponding bounding boxes $b_i$ and class labels $c_i$, we first obtain the binary mask $\mathbf{m} \in \{0, 1\}^{W \times H}$ indicating the bounding boxes of all objects as provided in the layout $\ell$. Next, we define a blur schedule function $s(t) : [0, T] \rightarrow [0, 1]$ to compute the current blur strength $s_t \in [0, 1]$ at training step $t \in [0, T]$. In its simplest form, it can be a linear mapping from training progress to blur strength $s(t) = 1 - t/T$, moving from strong blur ($s_0 = 1$) to clean images ($s_T = 0$). Given a start resolution of $W_0, H_0 \leq W_t, H_t \leq W, H$, we use $s_t$ to compute the intermediate image resolution at the current step $t$:

$$W_t = (1 - s_t) \cdot (W - W_0) + W_0 \tag{1}$$

$$H_t = (1 - s_t) \cdot (H - H_0) + H_0 \tag{2}$$

Using a bilinear image resizing operation $\psi(\mathbf{x}, w, h)$, we can then generate the low-resolution (LR) image $\mathbf{x}_{LR,t} \in \mathbb{R}^{W \times H \times C}$ for the current timestep $t$ by first downsampling $\mathbf{x}_{HR}$ to $W_t, H_t$ and subsequent upsampling to match the image resolution of $\mathbf{x}_{HR}$, thus removing high-frequency details while retaining structural content:

$$\begin{aligned} \mathbf{x}_{LR,t} &= \text{blur}(\mathbf{x}_{HR}, s_t) \\ &= \psi_{up}(\psi_{down}(\mathbf{x}_{HR}, W_t, H_t), W, H) \end{aligned} \tag{3}$$

We have two options to perform object-level blurring: blur the foreground objects or the background. To produce the former, we can cut-and-paste the object regions of $\mathbf{x}_{LR}$ into $\mathbf{x}_{HR}$ using the binary mask $\mathbf{m}$ to produce $\hat{\mathbf{x}}_{LR \rightarrow HR}$. Similarly, we can generate an alternative image with a blurred background by cut-and-pasting the object regions of $\mathbf{x}_{HR}$ into $\mathbf{x}_{LR}$ to get $\hat{\mathbf{x}}_{HR \rightarrow LR}$.

$$\begin{aligned} \text{blur objects:} &\quad \hat{\mathbf{x}}_{LR \rightarrow HR} = \mathbf{m} \odot \mathbf{x}_{LR} + \overline{\mathbf{m}} \odot \mathbf{x}_{HR} \\ \text{blur background:} &\quad \hat{\mathbf{x}}_{HR \rightarrow LR} = \mathbf{m} \odot \mathbf{x}_{HR} + \overline{\mathbf{m}} \odot \mathbf{x}_{LR} \end{aligned} \tag{4}$$

where $\overline{\mathbf{m}}$ denotes the inverted mask, and $\odot$ is the element-wise Hadamard product. To control how often we want to use an image with blurred objects as opposed to an image with blurred background, we define the probability $p_{obj}$ of blurring the objects as opposed to the background, and randomly choose whether to return $\hat{\mathbf{x}}_{HR \rightarrow LR}$ or $\hat{\mathbf{x}}_{LR \rightarrow HR}$ for the current sample.

**Table 1: Main results. We report the mean and standard deviation over three runs and mark the best mean in bold. Using our proposed ObjBlur schedule, we achieve better performance across global image FID and object-level SceneFID while often reducing the variance at the same time. Note: [43] uses a slightly different evaluation protocol; thus, the scores are not directly comparable to [10, 29].**

| Model | COCO | | | Visual Genome | | |
|---|---|---|---|---|---|---|
| | FID ↓ | SceneFID ↓ | DS ↑ | FID ↓ | SceneFID ↓ | DS ↑ |
| LostGAN [29] | 27.34 ± 0.88 | 13.73 ± 0.66 | 0.47 ± 0.08 | 31.53 ± 1.50 | 11.76 ± 0.46 | 0.46 ± 0.08 |
| **LostGAN [29] + ObjBlur** | **21.92 ± 0.88** | **11.41 ± 0.31** | **0.48 ± 0.09** | **28.72 ± 1.39** | **10.76 ± 0.45** | **0.47 ± 0.08** |
| CAL2IM [10] | 16.72 ± 0.85 | 8.22 ± 0.29 | **0.44 ± 0.09** | 20.41 ± 1.45 | 6.98 ± 1.34 | **0.39 ± 0.09** |
| **CAL2IM [10] + ObjBlur** | **15.40 ± 0.29** | **7.98 ± 0.14** | **0.44 ± 0.09** | **19.94 ± 0.94** | **6.85 ± 0.26** | **0.39 ± 0.09** |
| LayoutDiffusion [43] | 17.59 ± 0.09 | 7.26 ± 0.22 | **0.46 ± 0.09** | 15.50 ± 0.21 | 6.61 ± 0.12 | **0.45 ± 0.09** |
| **LayoutDiffusion [43] + ObjBlur** | **17.19 ± 0.15** | **4.76 ± 0.35** | **0.46 ± 0.09** | **14.55 ± 0.29** | **5.39 ± 0.19** | **0.45 ± 0.09** |

In other words, for each image sample, we either blur the objects as defined by the layout $\ell$ or the background before continuing with model training. The blur strength is progressively reduced, using the blur schedule function $s(t)$ and the current training step $t$, while the start resolution used to compute $\mathbf{x}_{LR}$ defines the initial blur strength. See Figure 2 for a visualization and Algorithm 1 for an algorithmic overview of our method. We analyze different blur schedules $s(t)$, start resolutions, object probability $p_{obj}$ and schedule durations in section 5.

In contrast to CutBlur [36], which performs fixed blurring of a random patch, our method consists of a semantic-driven curriculum. Through progressive object-level blurring, we do not induce spatial confusion, unrealistic patterns, or loss of semantic content in the training data. Our proposed ObjBlur focuses on object-level blurring and implicitly guides the model into respecting the input layout by learning the boundaries between blurred and non-blurred areas. Because the blurring schedule converges towards clean images, the data augmentation does not leak into the generations of the final model.

## 4 EXPERIMENT SETUP

### 4.1 Datasets

We use COCO-Stuff [3] and Visual Genome [17] and follow standard data filter procedures as in the corresponding works [10, 29, 43]. In COCO-Stuff, the annotations contain 80 thing classes (person, car, etc.) and 91 stuff classes (sky, road, etc.). Boxes that are smaller than 2% of the image area are eliminated, and we use images with 3 to 8 objects. Finally, images that belong to *crowd* are filtered, resulting in 74,777 train and 3,097 val images. In Visual Genome, we select object and relationship categories occurring at least 2000 and 500 times in the train set, respectively, choose images with 3 to 30 bounding boxes, and ignore all small objects, resulting in 62,565 train, 5,506 val, and 5,088 test images.

### 4.2 Evaluation Metrics

We choose multiple metrics to evaluate our model and compare it with baselines. To evaluate the quality and diversity of the images, we use FID [11]. To assess the visual quality of individual objects, we choose the SceneFID [30], which corresponds to the FID applied

on cropped objects as defined by the bounding boxes, and the classification accuracy score (CAS) [20], which measures how well an object classifier trained on generated image crops can perform on real image crops. To evaluate the diversity between images generated from the same layout, we adopt the procedure from LayoutDiffusion [43] and use LPIPS [38] as the diversity score (DS) between two images generated from the same layout (one-to-many mapping). As it is a distance metric, higher values indicate greater image diversity.

### 4.3 Models & Training Details

We choose two adversarial [10, 29] and one recent diffusion [43] models and use the official PyTorch implementations available on GitHub to evaluate our proposed ObjBlur schedule. For [29] and [10], we increase the batch size to 512 and train on 8 NVIDIA RTXA6000 GPUs to speed up training. Given this change, we also retrain the baseline, improving it compared to the reported performance as a side effect. We train for 200 epochs, which takes about two days. For [43], we leave everything unchanged and train on 8 NVIDIA A100 GPUs for 300k iterations with a batch size of 64, which takes about two days. The image resolution is set to 128×128. We train every model three times and report mean and standard deviation to evaluate training stability.

## 5 RESULTS

We first discuss our main results in Table 1 and then perform an extensive analysis on the effect of different hyperparameters using LostGAN [29] and the COCO [3] dataset as a simple baseline. Finally, we critically examine the importance of performing object-level blurring as opposed to full image blurring, random patch blurring (as in CutBlur [36]), and random mask blurring.

### 5.1 Quantitative Evaluation

Our main results are summarized in Table 1 and Table 2. ObjBlur significantly improves the performance of LostGAN [29] and CAL2IM [10] across both FID and SceneFID in terms of mean and standard deviation in three runs without any changes to the model architecture or optimization. We achieve a relative improvement of 19.82% in FID and 16.89% in SceneFID on COCO with [29]. With [10], our

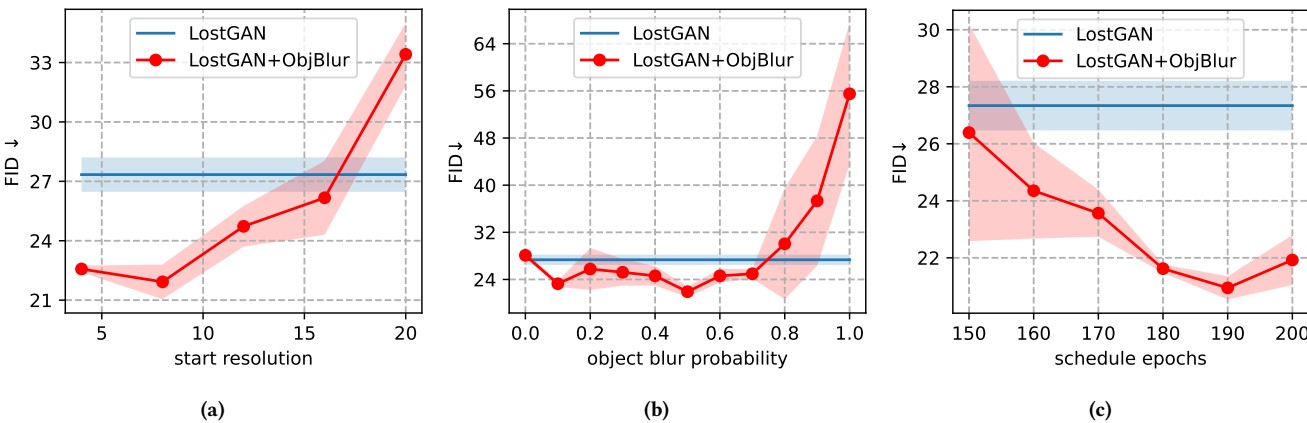

Figure 3: (a) We test different initial blurring strengths corresponding to the used start image resolution to compute $x_{LR}$ and find that 4 and 8 perform best. (b) We analyze the object blur probability $p_{obj}$, which defines the ratio between object vs. background blurring, and find that 50% works best. Blurring objects too often negatively affects performance. (c) We study the effect of schedule duration during which we apply our blurring schedule and find that 95% of training time, corresponding to 190 out of 200 epochs, performs best.

method significantly reduces the mean FID and SceneFID from 16.72 to 15.40 (a relative improvement of 7.89%) and from 8.22 to 7.98, respectively, while reducing the FID and SceneFID standard deviations from 0.85 to 0.29 and from 0.29 to 0.14. Our method also improves LayoutDiffusion [43], thus achieving a new state-of-the-art in terms of FID and SceneFID on both COCO and Visual Genome. Even though diffusion models are much more stable to train as compared to GANs, we still observe a significant improvement, decreasing the global image FID from 17.59 to 17.19 on COCO, and from 15.50 to 14.55 on Visual Genome. In particular, we improve the SceneFID by 34.43% on COCO, and 18.45% on Visual Genome. In terms of generated image diversity, we reach comparable or better DS scores, showing that ObjBlur maintains or improves sampling diversity. Table 2 shows that ObjBlur produces much more recognizable objects across all tested models and datasets, improving [29] on COCO by 2.44pps, and [43] on Visual Genome by 3.70pps.

## 5.2 Effect on Training Stability

To better understand the influence of our method, we analyze the performance during training of [29] with and without ObjBlur (Figure 1). In terms of FID, our method leads to a much smoother convergence with better final performance. Compared to the baseline, performance improves steadily throughout training time, and the standard deviation across runs is also much lower, especially at convergence. Therefore, our approach can be seen as an effective stabilization and regularization method. We also compare our method's capability to produce diverse images for the same layout using LPIPS as the diversity score and find that ObjBlur achieves comparable scores. In other words, our proposed schedule does not lead to a loss of sampling diversity.

## 5.3 Importance of Initial Blur Strength

The initial blurring strength (i.e., the resolution used to compute LR image regions) plays an important role and must be balanced to

Table 2: Classification accuracy scores (CAS) with and without ObjBlur (higher is better). We achieve consistently better scores with ObjBlur.

| Model | COCO | Visual Genome |
|---|---|---|
| real images | 51.04 | 48.07 |
| LostGAN [29] | 28.70 | 25.89 |
| **LostGAN [29] + ObjBlur** | **31.14** | **26.40** |
| CAL2IM [10] | 32.20 | 27.94 |
| **CAL2IM [10] + ObjBlur** | **32.43** | **28.09** |
| LayoutDiffusion [43] | 43.60 | 36.45 |
| **LayoutDiffusion [43] + ObjBlur** | **45.65** | **40.15** |

successfully regularize the training process. Starting with too much blurring could lead to overfitting on the boundaries, especially if a schedule function with a slow ramp-up is used. Starting with too little blurring could remove any potential benefits of using a CL schedule because the difference to training with clean images is too small. Furthermore, it could hinder training by changing the data distribution too quickly during the early training phase. Our results are summarized in Figure 3a. We find that starting with an LR resolution of 4 and 8 works best with steady degradation when starting higher.

## 5.4 Effect of Blur Objects/Background Ratio

We propose to select between the blurring of objects or the background at random defined by $p_{obj} = 0.5$. To better understand the impact of the ratio, we conduct additional experiments that range from always blurring the background $p_{obj} = 0.0$ to always blurring the objects $p_{obj} = 1.0$. Our results in Figure 3b show that many configurations lead to better performance, but 50% works best. Applying a blur schedule focusing on the background is generally

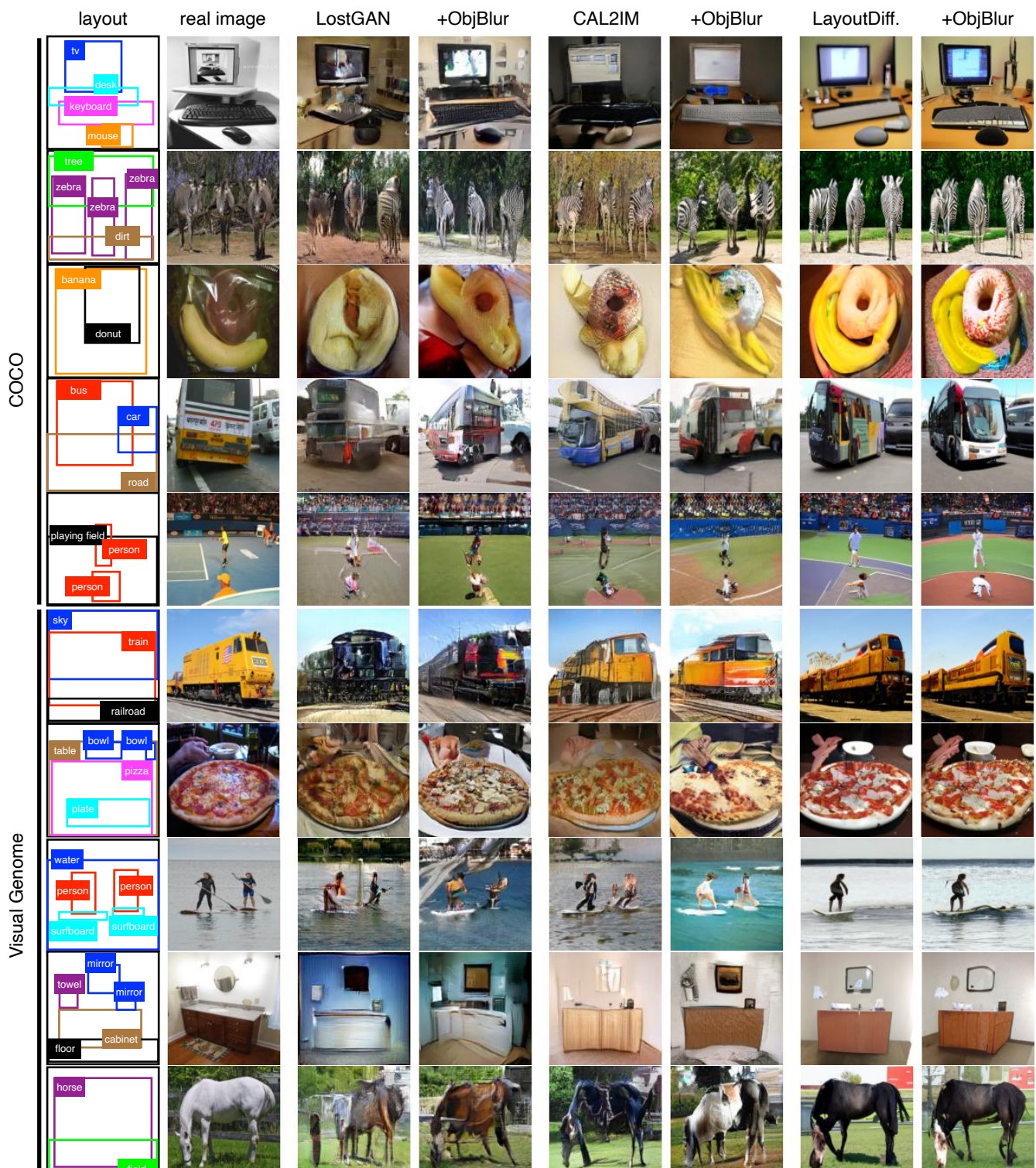

**Figure 4: Visual comparison of generated images with and without using ObjBlur during training. Our images are subjectively better, with more fine-grained details, better texture, more recognizable objects and higher global image coherence.**

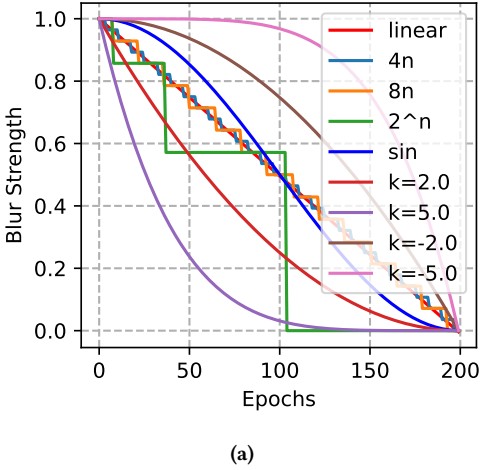

(a)

| schedule | FID ↓ | SceneFID ↓ |
|----------|-------|-----------|
| none | 27.34 | 13.73 |
| linear | 25.37 | 12.93 |
| 4n | 26.21 | 13.11 |
| 8n | 26.95 | 13.45 |
| 2^n | 23.80 | 12.20 |
| sin | **21.92** | **11.41** |
| k=2.0 | 33.60 | 16.57 |
| k=5.0 | 32.57 | 16.34 |
| k=-2.0 | 26.88 | 13.34 |
| k=-5.0 | 23.45 | 11.63 |

(b)

**Figure 5: We test different schedule functions** $s(t)$ **to compute the current blurring strength. (a) Visualization of different schedule functions. (b) Results: Many choices yield better performance compared to the baseline, but using a** sin **schedule function produces the best scores.**

more stable. We hypothesize that blurring objects too often negatively affects performance due to overfitting on low-resolution object boundaries.

### 5.5 Effect of Schedule Duration

The blurring schedule does not necessarily have to run for the entire duration of training. For example, it is possible to use the CL schedule within the first $n$ epochs and then fine-tune on clean images for the remaining iterations. We ablate the impact of using it within 70% to 100% of the training time, see Figure 3c and find that the model benefits from a short fine-tuning stage on clean images in the last 5% of epochs. Interestingly, the standard deviation across multiple runs increases towards longer fine-tuning phases on clean images, indicating problematic convergence behaviour. We hypothesize that this is due to shorter adaptation times during the CL schedule.

### 5.6 Visual Comparison

A comparison of generated images using baseline models and our method is shown in Figure 4 for adversarial approaches LostGAN [29], CAL2IM [10], and LayoutDiffusion [43]. When using ObjBlur, we find that the images are generally of similar quality. However, looking closer, one can see that our method often produces better images and more recognizable objects in comparison, especially on the GAN-based models. We provide four more pages of visual results in the appendix.

### 5.7 Effect of Different Schedule Functions

We ablate several other functions to compute the current blur strength, see Figure 5: linear, step functions with step sizes of 4 and 8, power of two steps with an exponential increase of steps to allow more time for adjustment and exponential functions with different rates. Although most achieve better results than the baseline, sin

performs significantly better than others, providing an initial warm-up and final fine-tuning along a symmetric transition throughout the schedule. Interestingly, an exponentially schedule function emphasizing a long warm-up performs second best, indicating the potential benefits of a kind of "pre-training" on low-resolution images. We leave further exploration to future work.

### 5.8 ObjBlur vs. CutBlur

To test the importance of our proposed object-level schedule, we compare it with a CutBlur [36] version of our CL schedule. We use the official implementation to select random patches and keep all other parameters, such as blur strength and schedule functions, constant. The results can be found in Table 3. Using CutBlur leads to degraded performance and yields a generator that produces blurred patches after training. Combining CutBlur with our schedule improves the baseline, indicating that a blurring schedule is effective. Our proposed ObjBlur, which applies blurring based on semantic information provided by the object annotations, is the best across both FID and SceneFID by a significant margin.

### 5.9 Effect of Object-Levelness

A critical question to investigate is whether the benefits of our proposed blurring schedule are due to semantic differentiation between foreground objects and background or if similar performance can be achieved by choosing the correct amount of blurring on a dataset level. To answer this question, we perform a random mask assignment experiment. We reuse the object-based masks computed with our proposed approach and shuffle the image-to-mask assignment during training. This will effectively keep the amount of blur on a dataset level constant and answer whether the semantic-driven masks are significant. Alternatively, we test whether applying the blur on the entire image is beneficial. Different blur techniques are visualized in Figure 6. Our results in Table 3 indicate that a

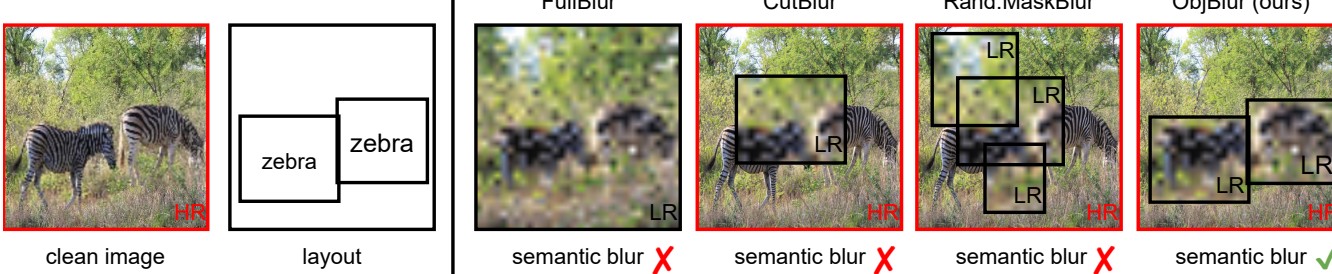

**Figure 6: Visualization of different blurring techniques. We compare our object-level blurring against blurring the full image (FullBlur), random patch blurring (CutBlur [36]), and a version in which we shuffle the image-mask alignment (Rand.MaskBlur). Results in Table 3.**

semantic-driven blurring schedule is critical and that performance suffers whenever objects are inconsistently blurred.

## 6 LIMITATIONS & FUTURE WORK

While our method can stabilize training and improve performance, there are a few limitations and possibilities for future work. We exclusively studied blurring as an augmentation function. The potential applicability of other augmentations, such as additive noise, remains unexplored. Uniform blurring across all images and object classes ignores object and sample difficulty differences. Investigating a dynamic blurring schedule for individual objects, object classes, or specific samples could benefit future research. Although our findings underscore the necessity of object masks, we also demonstrate that combining CutBlur [36] with our CL schedule surpasses baseline performance. Further studies are needed to determine whether similar performance can be achieved using masks without requiring object annotations. Combining ObjBlur with blurring diffusion models such as in [6, 12] would be interesting. Finally, we ask whether our approach could work on single-object datasets such as ImageNet by blurring important (object-centric) areas and whether it benefits low-data regime scenarios.

## 7 REPRODUCIBILITY & ETHICS STATEMENT

Our method can easily be reproduced as it only requires changes to the data loader of existing layout-to-image models to apply a progressive object-level blur using standard down- and upsampling operations. We show an example implementation in the appendix. Our method is plug-and-play and improves existing layout-to-image generation models. As such, it inherits risks such as being misused to spread fake news, invading privacy, and potential copyright issues due to using real-world datasets. To counter these issues, it's important to develop advanced deepfake detection technologies and enforce ethical guidelines to differentiate synthetic images from real ones, ensuring responsible use of the technology.

## 8 CONCLUSION

In this work, we introduced ObjBlur, an innovative curriculum learning strategy based on object-level blurring that significantly improved layout-to-image generation models. Our approach reaches

**Table 3: Combining CutBlur [36] with our proposed CL schedule performs well, but applying the CL schedule on objects instead of random patches yields the best performance suggesting that semantic-driven blur masks are important.**

| blur technique | CL | FID ↓ | SceneFID ↓ |
|---|---|---|---|
| none | ✘ | 27.34 | 13.73 |
| CutBlur [36] | ✘ | 57.53 | 31.51 |
| CutBlur [36] | ✔ | 23.94 | 12.05 |
| FullBlur | ✔ | 27.26 | 13.81 |
| Rand.MaskBlur | ✔ | 26.68 | 12.80 |
| **ObjBlur** | ✔ | **21.92** | **11.41** |

state-of-the-art performance, better training stability, and reduced variance across different runs through a systematic progression from strong blurring to progressively cleaner images during training. ObjBlur is plug-and-play, and only requires modifications to the data loader, which makes it easy to utilize. Its compatibility with generative adversarial networks and diffusion models underscores its versatility in various generative modelling paradigms. Our research explores curriculum learning in the context of layout-to-image generation for the first time, and we hope it leads to further investigations into the potential of curriculum learning and data augmentation within generative models.

## ACKNOWLEDGMENTS

Will be added later.

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
