# OpenReview forum: "ObjBlur: A Curriculum Learning Approach With Progressive Object-Level Blurring for Improved Layout-to-Image Generation"
_acmmm.org/ACMMM/2024/Conference — MM2024 Poster_

### Official Review · Reviewer_CVZd · 2024-05-24

**Rating:** 4
**Confidence:** 2

**Summary:**

This paper proposes a novel curriculum learning approach designed to enhance layout-to-image generation models by progressively applying object-level blurring during training. The method leverages blurring as an intuitive way to adjust task difficulty, maintaining structural content while progressively introducing high-frequency details to the model. This approach stabilizes training, improves image quality, and smooths convergence by gradually transitioning from strong blurring to clearer images. The effectiveness of the proposed method is demonstrated across various datasets.

**Strengths:**

1. The paper is well-organized and clearly written.
2. The proposed method is both novel and effective.
3. Extensive experiments validate the method's effectiveness in diverse settings.
4. The visualization results are impressive.
5. The limitations of the method are thoroughly discussed.

**Limitations:**

1. The authors should incorporate error analyses, such as confusion matrices, to provide a deeper understanding of the proposed method's performance.
2. Results for different hyper-parameters (hyper-parameter sensitivity analyses) should be provided to evaluate the robustness of the method.
3. The proposed method applies a uniform blurring strategy across all images and object classes, which does not account for the varying difficulty levels of different objects and scenes.
4. The method relies heavily on the availability of accurate object annotations to generate the blurring masks. This dependency could limit the applicability of the approach to datasets where such detailed annotations are not available.
5. The progressive blurring technique adds additional computational steps during the data loading process, which could increase the overall training time.

**Suitability:**

2

---

### Official Review · Reviewer_obap · 2024-05-24

**Rating:** 4
**Confidence:** 3

**Summary:**

The paper mainly uses the Curriculum Learning method to alleviate the difficulty of the layout-to-image task and improve the model training effect. It can be applied to existing methods in a plug-and-play manner. The main method is to guide the model to learn layout-to-image generation from easy to difficult by blurring images with different strength, and by randomly selecting the foreground or background of the blur, the model focuses on layout information by learning the boundary between the foreground and background. Experiments show the gain effect on GAN and Diffusion Models after using this method, and extensive ablation experiments are conducted.

**Strengths:**

1. This method is the first to use Curriculum Learning for layout to image tasks, optimizing the learning process; In addition, to enhance the processing of layout information, a random blur foreground and background method is adopted. Overall, it is simple and effective, with innovation
2. The experimental part applied this method on GAN and Diffusion Models, demonstrating its efficiency in image quality and diversity. Sufficient ablation experiments were conducted to demonstrate the effectiveness of the method in training stability, initial blur intensity selection, blur ratio and schedule design.

**Limitations:**

1. This paper designs a random blurring method to optimize layout learning. Although they explains that its main mechanism is to enhance the model's attention to boundaries between the foreground and background, there is no in-depth and detailed analysis on how this method is beneficial for the model to process layout information.
2. Although this method performs significantly on GAN models, its gain on Diffusion Models is relatively not significant, while in many metrics (such as SceneFID), DM performs the best as a baseline (possibly related to its training method for denoising learning)

**Suitability:**

3

---

### Official Review · Reviewer_jdaG · 2024-05-25

**Rating:** 3
**Confidence:** 2

**Summary:**

This paper introduces a novel method called ObjBlur, which leverages curriculum learning through progressive object-level blurring to enhance the quality and stability of layout-to-image generation models. This approach significantly improves image quality, training stability, and achieves state-of-the-art results on complex datasets in COCO and Visual Genome.

**Strengths:**

The use of progressive object-level blurring as a curriculum learning strategy is a novel and effective method to improve training stability and image quality in layout-to-image generation models. ObjBlur demonstrates significant improvements in image generation quality and training stability across various models and datasets, achieving state-of-the-art results on COCO and Visual Genome datasets.

**Limitations:**

Strong initial blurring could lead to overfitting on the boundaries of objects, especially if the transition to clean images is not well-balanced.
The method assumes that progressively reducing blur uniformly increases task complexity in a manner beneficial for all types of images and objects. However, different objects and scenes may have varying inherent complexities that are not adequately addressed by a uniform blurring schedule.
The effectiveness of ObjBlur hinges on the availability of accurate object annotations. In real-world scenarios, obtaining precise annotations can be challenging, and any inaccuracies can negatively impact the training process. This dependency limits the method's applicability to datasets where high-quality annotations are not available or feasible to obtain.
While the method shows improved performance on specific datasets, its ability to generalize to unseen data distributions is theoretically uncertain. The training improvements observed might be specific to the characteristics of the datasets used (e.g., COCO and Visual Genome) and may not translate to datasets with different properties.

**Suitability:**

2

---

### Meta-Review · Area_Chair_Kq7V · 2024-06-30

**Recommendation:** Accept (Poster)
**Confidence:** 5

**Metareview:**

This paper receives two borderline accept and one borderline reject. Reviewers acknowledge the novelty of the paper but raises several questions regarding the method design and experiments. Authors addressed the questions during rebuttal. After reading the paper, reviews, and rebuttal, the area chair decides to recommend acceptance and suggest the authors to take the reviewers' comments into consideration for the camera-ready version.